# Development of Large Hollow Particles for Pulmonary Delivery of Cyclosporine A

**DOI:** 10.3390/pharmaceutics15092204

**Published:** 2023-08-25

**Authors:** Yongpeng Huang, Hui Tang, Xiangyan Meng, Zitong Zhao, Yanli Liu, Dongxin Liu, Bo Chen, Zhiyun Zou

**Affiliations:** State Key Laboratory of NBC Protection for Civilian, Beijing 102205, China; h380645362@163.com (Y.H.); fannyth@163.com (H.T.); mxy_123@sohu.com (X.M.); zzt4112@163.com (Z.Z.); liuh3062023@163.com (Y.L.); liudongxin95@163.com (D.L.)

**Keywords:** dry powder inhalations, large hollow particles, hydroxypropyl methylcellulose, spray drying, pulmonary delivery, cyclosporine A

## Abstract

The purpose of this study was to prepare large hollow particles (LHPs) by spray drying for pulmonary delivery of cyclosporine A (CsA), using L-Leucine (LEU) and hydroxypropyl methylcellulose (HPMC) as excipients and ammonium bicarbonate (AB) as a porogen. The prepared LHPs were spherical particles composed of both CsA and LEU on the surface and HPMC on the inner layer. The formulation of CsA-LEU-0.8HPMC-AB as typical LHPs showed excellent in vitro aerodynamic performance with a minimum mass median aerodynamic diameter (MMAD) of 1.15 μm. The solubility of CsA-LEU-0.8HPMC-AB was about 5.5-fold higher than that of raw CsA, and the dissolution of CsA-LEU-0.8HPMC-AB suggested that the drug was released within 1 h. The cell viability of the A549 cell line showed that CsA-LEU-0.8HPMC-AB was safe for delivering CsA to the lungs. In addition, inhalation administration of CsA-LEU-0.8HPMC-AB with the *C*_max_ and AUC_0–∞_ increasing by about 2-fold and 2.8-fold compared with the oral administration of Neoral^®^ could achieve therapeutic drug concentrations with lower systemic exposure and significantly improve the in vivo bioavailability of CsA. From these findings, the LHPs, with the advantage of avoiding alveolar macrophage clearance, could be a viable choice for delivering CsA by inhalation administration relative to oral administration.

## 1. Introduction

Cyclosporine A (CsA) is one of the most important transplantation drugs. It was discovered and isolated from the fungus in 1970 and proved to have immunosuppressive characteristics in 1976 [1]. CsA is a milestone in organ transplantation [2] and shows a variety of biological activities, such as anti-fungal, anti-inflammatory, and anti-parasitic properties [3]. In addition, clinical studies indicate that CsA has strong therapeutic potential for chronic asthma and airway inflammation [4]. However, the clinical application of CsA is partly limited due to its low oral bioavailability, which is related to poor solubility, low intestinal permeability, and CYP3A-related biotransformation [5]. In addition, systemic overexposure to CsA could cause a series of adverse effects, such as nephrotoxicity, hepatotoxicity, hypertension, nausea, and vomiting [6]. Therefore, developing CsA formulations with high bioavailability is the current research focus.

Pulmonary drug delivery is a newly developed drug delivery method that has the advantages of non-invasive drug delivery, high blood concentration, rapid absorption, and no first-pass hepatic metabolism [7,8]. Dry powder inhalers (DPIs) exhibit the advantages of being a light device with no propellant, having good patient compliance, easy long-term storage of drugs, and being an advanced preparation for pulmonary drug delivery [9]. The deposition mechanisms of DPIs in pulmonary drug delivery mainly include inertial impaction, sedimentation, and Brownian diffusion and are determined by the aerodynamic diameter of the particles [10]. In pulmonary drug delivery, particles with an aerodynamic diameter in the range of 1–5 µm are generally deposited deeply into the lungs by sedimentation. Particles with an aerodynamic diameter greater than 5 µm are mainly deposited in extrathoracic areas by inertial impaction, while those with an aerodynamic diameter less than 0.5 µm are exhaled out of the body by Brownian diffusion [11]. Nevertheless, one of the clearance mechanisms in pulmonary drug delivery is phagocytosis by alveolar macrophages in the lungs, which can clear particles with geometric diameters of 1–2 µm [12,13,14]. 

According to Equation (1) [15], one of the effective methods to prevent phagocytosis by alveolar macrophages in pulmonary drug delivery is to prepare inhalable DPIs with a large geometric diameter and low density. Large porous particles (LPPs) or large hollow particles (LHPs) are the types of DPIs that meet these conditions. LPPs/LHPs are characterized by geometric sizes in the 5–30 µm range and mass densities lower than 0.4 g/cm^3^ [10,12]. Another major advantage of LPPs/LHPs is the better aerosolization efficiency of the DPIs compared to non-porous particles due to their low density, larger size, and consequently lower tendency for aggregation [16,17]. Therefore, preparing LPPs/LHPs with stable physicochemical properties may be the challenge of pharmaceutical technology in pulmonary drug delivery.
(1)$$d_{a}=d_{g}\times \sqrt{\frac{\rho }{\chi }}$$
where *d*_a_ is the aerodynamic diameter, *d*_g_ is the geometric diameter, *ρ* is the density, and *χ* is the dynamic shape factor. 

Spray drying is a mature technique in the pharmaceutical industry and is also used to produce LPPs/LHPs [18]. L-Leucine (LEU) is one of the most commonly used excipients in the preparation of inhalable powders by spray drying, which has the functions of changing the particle morphology [19,20], and improving in vitro particle deposition [21]. Hydroxypropylmethyl cellulose (HPMC) is one of the most commonly used physical property modifiers in the pharmaceutical industry due to its unique solubility, low surface tension, reversible sol–gel conversion with temperature, low hygroscopicity, high glass transition temperature, excellent plastic deformation, and compactness [22]. In addition, HPMC also has the characteristics of improving the dissolution, stability, and bioavailability of drugs [23,24,25]. Ammonium bicarbonate (AB) is one of the commonly used porogens for preparing LPPs/LHPs by spray drying, and it can be decomposed into ammonia and carbon dioxide gas bubbles to form porous or hollow particles during spray drying processes [26,27].

In this study, we aimed to prepare LHPs by spray drying for pulmonary delivery of CsA, using LEU and HPMC as excipients and AB as a porogen. The compatibility between the drug, HPMC, and LEU was verified using Fourier transform infrared spectroscopy (FTIR), powder X-ray diffraction (PXRD), differential scanning calorimetry (DSC), and thermogravimetric analysis (TGA). The morphology, particle size distribution, density, and flowability of LHPs were examined to characterize the physicochemical performance of DPIs. X-ray photoelectron spectroscopy (XPS) was used to investigate the surface characteristics of LHPs. Moreover, we expanded in vitro aerosolization performance and dissolution to achieve a more accurate characterization of LHPs. Finally, we evaluated the in vivo pharmacokinetics in rats to clarify the advantage of inhalation administration of LHPs. To our best knowledge, using LEU and HPMC as excipients to prepare inhalable LHPs for pulmonary delivery of CsA has not been reported so far.

## 2. Materials and Methods

### 2.1. Materials

Hydroxypropyl methylcellulose (HPMC, E3), ammonium bicarbonate (AB, AR), and ethanol (absolute, ≥99.5%) were purchased from Macklin (Shanghai, China). L-Leucine (LEU, ≥99.0%) was purchased from Energy Chemical (Shanghai, China). Acetonitrile (HPLC grade) was obtained from Merck KGaA (Darmstadt, Germany). Cyclosporine A (CsA, USP grade) was provided by BioDuly (Nanjing, China). Water was purified using a Milli-Q purification system (Millipore, Darmstadt, Germany).

### 2.2. Preparation of Large Hollow Particles

The LHPs were produced by spray drying using a B-290 spray dryer (Büchi, Switzerland) with a 2.0 mm two-fluid nozzle, and the spray dryer was operated in closed-loop blowing mode with the application of a B-295 inert loop (Büchi, Switzerland). Firstly, CsA was ultrasonically dissolved in a mixture of ethanol and water with a volume ratio of 50:50 at a concentration of 2.0 mg/mL. Secondly, LEU was ultrasonically dissolved in the CsA solution to reach a concentration of 0.4 mg/mL. Thirdly, HPMC was ultrasonically dissolved in the above solution to obtain concentrations of 0.4, 0.8, 1.2, 1.6, or 2.0 mg/mL. In some cases, AB as a porogen was added at a concentration of 2.0 mg/mL under ultrasound before spray drying to minimize its decomposition. The formulations are shown in Table 1. The parameter settings of B-290 were as follows: 170 °C inlet temperature, 20% feed pump rate, 100% aspirator rate, and 414 L/h gas flow rate.

### 2.3. Identification of Powders and Drug Content

In order to investigate the characteristics of the CsA in the formulations, Fourier transform infrared spectroscopy (FTIR) was applied by the KBr pellet method using a Nicolet IS 10 apparatus (Madison, WI, USA). The samples were scanned at a resolution of 0.4 cm^−1^ in the wavenumber region of 4000–400 cm^−1^. 

The content (%) of CsA was measured by dissolving ~25 mg formulation in 25 mL of ethanol and water with a volume ratio of 50:50, and CsA content was quantified by the HPLC method with slight modifications, according to the literature (Jiang et al., 2022). Briefly, the Agilent 1260 Infinity Ⅱ HPLC system (Palo Alto, CA, USA) was used, and the ZORBAX 300 SB C8 column (250 mm × 4.6 mm, 5.0 μm) was selected for separation of CsA at 60 °C. The mobile phases were acetonitrile/water (60:40, *v*/*v*), and the flow rate was 1.0 mL/min. The injection volume was 10 μL, and the detection wavelengths were 205 nm.

### 2.4. Morphology

Scanning electron microscopy (SEM) was used to characterize the morphology of DPIs using a Hitachi SU8020 apparatus (Tokyo, Japan) at an acceleration voltage of 5 kV. All samples were attached to specimen stubs by double-sided adhesive tape and sputter-coated with a gold layer using a Hitachi MC1000 ion sputter (Tokyo, Japan). 

### 2.5. Particle Size Distribution

The geometric diameter of the DPIs was measured using a Mastersizer 3000 equipped with aero dispersion units (Malvern Panalytical Ltd., Malvern, Britain) at a dispersing pressure of 2 bar. The particle size distribution was expressed in terms of the particle median diameter at 90% of the volume distribution (D_90_).

### 2.6. Density and Flowability

The density of DPIs was measured using a 5 mL glass cylinder. The mass of the blank cylinder was recorded as *m*_0_, and the total mass of the cylinder filled with powder (the height of the powder corresponded to 5 mL) was recorded as *m*_1_. The cylinder was tapped from a height of 5 cm until the powder volume remained unchanged, and the final volume was recorded as *V*. The bulk density (*ρ_b_*), tap density (*ρ_t_*), and Carr’s index were calculated using the following equations:(2)$$\mathrm{Bulk}\;\mathrm{density}\;(\rho_{b})=\frac{m_{1}-m_{0}}{5}$$
(3)$$\mathrm{Tap}\;\mathrm{density}\;(\rho_{t})=\frac{m_{1}-m_{0}}{V}$$
(4)$$\mathrm{Carr}’s\;\mathrm{index}=\frac{\rho_{t}-\rho_{b}}{\rho_{t}}$$

### 2.7. Component Distribution on the Particle Surface

X-ray photoelectron spectroscopy (XPS) (Thermo ESCALAB 250Xi, Waltham, MA, USA) equipped with an aluminium Kα X-ray (1486.69 eV) source was used to investigate the surface characterization of DPIs. All powder samples were loaded on a sample holder, and wide, low-resolution survey spectra were recorded to identify the elements on the particle surface.

### 2.8. Powder X-ray Diffraction

Powder X-ray diffraction spectra (PXRD) were measured using a Bruker D8 Advance X-ray powder diffractometer (Germany) with Cu-Kα radiation (1.5406 Å) at a tube voltage of 40 kV and a tube current of 40 mA to analyze the character of DPIs. The samples were measured from 3 to 40° 2θ with a scanning rate of 2°/min and a step size of 0.02°.

### 2.9. Differential Scanning Calorimetry and Thermogravimetric Analysis

The thermal properties of DPIs were analyzed using differential scanning calorimetry (DSC) (Q 100, TA Instruments, Newcastle, DE, USA) and thermogravimetric analysis (TGA) (Q 500, TA Instruments, Newcastle, DE, USA). For DSC, samples were placed in hermetically sealed aluminum pan and analyzed at a nitrogen flow rate of 20 mL/min. The analysis was conducted from 40 °C to 320 °C at a heat rate of 10 °C/min. For TGA, samples were loaded on an open aluminium pan and heated from 25 °C to 600 °C at a scan rate of 10 °C/min under a nitrogen flow of 60 mL/min.

### 2.10. In Vitro Aerosolization Performance

According to the USP Chapter <601> specification for aerosols, the in vitro aerosolization performance of the DPIs was measured using a Next Generation Impactor (NGI, Copley Scientific Ltd., Nottingham, UK) equipped with a critical flow controller (TPK 100i-R) and a high-capacity pump (HCP6). Before each run, all NGI collection cups were covered with silicone oil in order to minimize particle bouncing, and the pre-separator was filled with 15 mL of diluent consisting of ethanol and water with a volume ratio of 50:50 at the same time. The air flow rate was set at 90 L/min and confirmed with a Copley flow meter (DFM 2000). The dry powder inhalation device with the powders (~10 mg) in the capsule was activated for 2.7 s into the NGI at a flow rate of 90 L/min. The powders remaining in the capsule, inhaler, induction port, and all NGI stages were rinsed with the diluent mentioned above, and the concentration of CsA in the diluent was assayed by the HPLC method mentioned in Section 2.3. 

The emitted fraction (EF) was calculated as the percentage of the drug found in the adaptor, induction port, pre-separator, and all impactor stages of the total recovered amount of drug. The fine particle fraction (FPF) and mass median aerodynamic diameter (MMAD) were calculated using the Inhalytix software (Version 1.0) (Copley Scientific Ltd., Nottingham, UK). FPF was defined as the fraction of the emitted dose with aerodynamic particle diameters lower than 5 μm. 

### 2.11. Solubility Test

The solubility test was carried out in pure water. The excess mass of raw CsA and formulation was added to the media and stirred with a magnetic stirrer (MR Hei-Tec, Heidolph, Germany) at 500 r/min and 25 °C for 24 h. Then, the samples were filtered by a 0.22 μm microporous membrane, and the filtrate was used to determine the concentration of CsA by the HPLC method, as described in Section 2.3.

### 2.12. In Vitro Dissolution

In vitro dissolution was carried out in simulated lung fluid. Simulated lung fluid was prepared with some modifications according to the literature [28]. The composition of simulated lung fluid included NaCl (6.40 g/L), CaCl_2_·2H_2_O (0.26 g/L), Na_2_HPO_4_ (0.15 g/L), NaHCO_3_ (2.70 g/L), NH_4_Cl (0.12 g/L), MgCl_2_ (0.10 g/L), Na_2_SO_4_·10H_2_O (0.18 g/L), Na_3_citrate·2H_2_O (0.16 g/L), Glycine (0.19 g/L), and dipalmitoylphosphatidylcholine (0.01% *w*/*v*). Raw CsA and formulation (ca. 2 mg CsA) were dispersed in 100 mL dissolution medium and placed in a magnetic stirrer (MR Hei-Tec, Heidolph, Germany) maintained at 100 r/min and 37 °C. At predetermined intervals, suspensions (0.5 mL) were withdrawn and centrifuged. The supernatants were diluted with 0.5 mL of ethanol and determined by the HPLC method, as described in Section 2.3.

### 2.13. In Vitro Cytotoxicity Assay

In vitro cytotoxicity was evaluated on A549 cells using the MTS (3-(4,5-dimethylthiazol-2-yl)-5-(3-carboxymethoxyphenyl)-2-(4-sulfophenyl)-2H-tetrazolium) assay in 96-well cell culture microplates. Briefly, A549 cells were grown in DMEM medium supplemented with 10% (*v*/*v*) fetal bovine serum (Gibco, Waltham, MA, USA), penicillin (100 U/mL), and streptomycin (100 μg/mL) (Beyotime Biotechnology, China). Cells were incubated at 100% relative humidity and 5% CO_2_ at 37 °C for 24 h and seeded into 96-well plates at a density of 2000 cells/well. Then, the sample suspensions were added to the wells and incubated for 24 h. Finally, 20 μL of MTS solution was added to each well and incubated for 1 h. The absorbance was measured at 490 nm (reference wavelength at 630 nm) using a microplate reader (Multiskan GO, Thermo Fisher, Waltham, MA, USA). Non-treated cells were used as a control sample, and the cell viability was calculated using the following equation: (5)$$\mathrm{Cell}\;\mathrm{viability}\;(\%)=\frac{{\mathrm{OD}}_{\exp}-{\mathrm{OD}}_{\mathrm{blank}}}{{\mathrm{OD}}_{\mathrm{control}}-{\mathrm{OD}}_{\mathrm{blank}}}\times 100\%$$
where OD_exp_, OD_blank_, and OD_control_ are the absorbance of the sample, the absorbance of the blank, and the absorbance of the control sample, respectively.

### 2.14. In Vivo Pharmacokinetic Studies

#### 2.14.1. Animals

Male Sprague Dawley rats with jugular vein intubation (250 ± 30 g) (Skillsmodel Animal Research Technology, China) were housed three per cage with a standard diet and water in the laboratory under 25 °C, 55% humidity, and a 12 h dark/light cycle. The rats were fasted but given water for 12 h before experiments. 

#### 2.14.2. Pharmacokinetic Studies

Twelve rats were randomly divided into two groups and slightly anesthetized using isoflurane before administration. The commercial oral formulation, Neoral^®^, was delivered to the stomach by oral administration, and the powders were delivered to the lungs using a dry insufflator powder delivery device (Huironghe Technology Co., Ltd., Beijing, China). The dose of oral administration and inhalation administration was 0.4 mg CsA/kg in rats according to the literature with slight modifications [29]. After administration, 200 μL of blood sample was withdrawn into a EDTA centrifuge tube from jugular vein intubation of rats at predetermined intervals. The blood samples were prepared and measured by the HPLC-MS/MS method according to the literature with slight modifications [30]. Briefly, 50 μL of the EDTA blood sample was spiked with 350 μL of methanol, 50 μL of internal standard cyclosporine D (CsD) and 50 μL of 0.5 mol/L ZnSO_4_ and shocked for 30 s. Then, the sample was kept at −20 °C for 30 min to precipitated proteins. After centrifugation at 14,000 r/min for 10 min, the supernatants were filtered through a 0.45 μm filter and analyzed by the established HPLC-MS/MS method. The content of CsA in the sample was determined with a Agilent 1290-6470 Triple Quad (USA). Chromatography was performed on a ZORBAX 300 SB C8 column (100 mm × 2.1 mm, 1.8 μm) at 60 °C. Acetonitrile (A) and 10 mmol/L ammonium formate + 0.1% formic acid (B) were selected as the mobile phases to separate CsA at the flow rate of 0.5 mL/min. The gradient condition was set as follows: 0–3 min, 50%A; 3–3.1 min, 50%A–70%A; 3.1–5 min, 70%A; 5–5.1 min, 70%A–50%A; 5.1–7 min, 50%A. The injection volume was 5 μL. Positive electrospray ionization was applied with transitions of *m*/*z* 602.5 → *m*/*z* 156.2 for CsA and *m*/*z* 609.5 → *m*/*z* 156.2 for the internal standard CsD in the multiple reactions monitoring (MRM) mode. 

### 2.15. Statistical Analysis

All experiment results were conducted in at least three independent experiments, unless otherwise specified, and data were expressed as the mean ± standard deviation (SD). Statistical variations between groups were calculated by the *t*-test. The level of *p* < 0.05 was considered statistically significant.

## 3. Results and Discussion

### 3.1. Identification of Powders and Drug Content in the Formulations

The FTIR results of raw CsA, LEU, HPMC, and CsA-LEU and CsA-LEU-0.8HPMC-AB formulations are shown in Figure 1. Raw CsA showed a band at 3313 and 2957 cm^−1^ for the N−H stretching vibration and the C−H stretching vibration, respectively, and the bands at 1621 and 1093 cm^−1^ were attributed to the C=O stretching vibration and the C−O stretching vibration, respectively [31]. LEU showed a wide band from 2400 to 3500 cm^−1^, indicating the presence of carboxyl group. The peaks at 1510 and 1573 cm^−1^ were attributed to the antisymmetric stretching vibration of O−C=O, and the peak at 1404 cm^−1^ was attributed to the symmetric stretching vibration of O−C=O. HPMC showed a band at 3450 and 2870 cm^−1^ for the O−H stretching vibration and the C−H stretching vibration, respectively, and the peak at 1049 cm^−1^ was attributed to the C−O stretching vibration [32,33]. Compared with CsA, LEU, and HPMC, the corresponding spectral bands of CsA-LEU and CsA-LEU-0.8HPMC-AB all existed with no significant shift, indicating that there was no significant interaction between CsA, LEU, and HPMC during the spray drying process.

Drug contents in the formulations are shown in Table 2. The CsA contents in the formulations were close to the theoretical contents, and the addition of AB in the formulations had almost no effect on the drug content, indicating that AB was completely decomposed into carbon dioxide, ammonia, and water during the spray drying process [34]. 

### 3.2. SEM Morphological Analysis

As shown in Figure 2, CsA-SD and CsA-LEU all exhibited a slightly rough surface and no hollow spherical shape, and the particle size was less than 5 µm. Interestingly, the presence of LEU did not alter the morphology of the particles, which was not entirely consistent with other reports [21,34] and may be due to the fact that LEU did not form a shell on the surface of the droplet during the spray drying process. CsA-LEU-0.4HPMC and CsA-LEU-0.4HPMC-AB with 0.4 mg/mL HPMC content maintained a similar spherical shape, and some particles showed a hollow structure, especially after adding AB. The formulations (E–L) with 0.8–2.0 mg/mL HPMC content exhibited a smooth surface and hollow spherical particles with a particle size of about 10 µm, indicating that the typical LHPs were successfully prepared and were expected to exhibit a low density. Furthermore, the addition of AB to formulations had no visible impact on particle morphology. When the particles of the LHPs were spherical, van der Waals forces and the aggregation tendency were at minimum levels, indicating good aerosol performance of the LHPs [35]. 

### 3.3. Particle Size Distribution

As shown in Table 2 (Appendix A), the particle size distributions of different formulations were different under the same spray drying parameters. The geometric diameter of CsA-SD and CsA-LEU was around 4.51 µm and 4.88 µm, respectively, indicating that adding 0.4 mg/mL LEU to the formulation did not obviously increase the geometric size of the particles, and it was similar to the results of other drugs reported in the literature [36,37]. However, the geometric diameter significantly increased (from 8.37 µm to 12.43 µm) when adding HPMC (from 0.4 mg/mL to 2.0 mg/mL) to the formulations, which may be due to an increase in viscosity of the formulation solution, leading to a larger droplet during the spray drying process, thus resulting in a larger particle size [38]. In addition, the geometric diameter would further increase by adding AB to the formulations, which was due to the blowing effect caused by the decomposition of AB into carbon dioxide and ammonia during the spray drying process [34,39]. 

### 3.4. Density and Flowability 

As shown in Table 2, the LHPs had lower bulk and tap densities compared to the non-hollow particles, as expected. CsA-LEU with 0.4 mg/mL LEU showed larger bulk and tap densities than CsA-SD, indicating that the presence of LEU increased the density of powders. However, the bulk and tap densities significantly decreased upon the addition of HPMC to the formulations compared with CsA-SD and CsA-LEU. Furthermore, the bulk and tap densities of the formulation with AB would further decrease compared to formulations containing the same concentration of CsA, LEU, and HPMC, which was related to the decomposition of AB during the drying process [37,40]. The typical LHPs had a bulk density of around 30.40–77.80 mg/cm^3^ and a tap density of around 42.03–111.58 mg/cm^3^. In addition, for the typical LHPs with or without AB, the bulk and tap densities were positively correlated with HPMC concentrations in the range of 0.8 to 2.0 mg/mL. The formulation of CsA-LEU-0.8HPMC-AB had the smallest bulk and tap densities, with values of 30.40 ± 2.16 mg/cm^3^ and 42.03 ± 2.14 mg/cm^3^, respectively. Due to its low density and larger size, CsA-LEU-0.8HPMC-AB was expected to have better aerosolization efficiency [16].

Carr’s index was commonly used to indicate the flowability of dry powders [41,42], with values of <25 suggesting good flowability and >40 indicating poor flowability. Carr’s index values of CsA-SD and CsA-LEU were all >45, indicating poor flowability, and the values of the formulations with HPMC were in the range of 28–32, indicating an improvement in flowability. In addition, the Carr’s index value of the formulation with AB was slightly smaller than that of the formulation containing the same concentration of CsA, LEU, and HPMC, indicating that AB further improved the flowability of dry powders. The formulation of CsA-LEU-0.8HPMC-AB with the smallest Carr’s index value of 27.73 ± 1.68 showed acceptable flow properties.

### 3.5. Particle Surface Composition Analysis

XPS was used as a surface quantitative analytical technique to provide information about the molar ratios of components at the surface of particles with a probing depth of 2–10 nm [43]. The XPS spectra and surface composition of the LHPs are shown in Figure 3 and Table 3. The carbon (C) atomic concentrations of CsA, LEU, and HPMC were higher than those of the stoichiometric values, which is a common phenomenon in XPS analysis of organic compounds due to the presence of adventitious carbon overlayers on the surface consisting of C–H bonding [44,45]. The C and nitrogen (N) compositions of CsA-LEU were lower than those of CsA, while the oxygen (O) composition was higher; this was exactly the opposite compared to LEU, indicating that the surface of CsA-LEU may be composed of both CsA and LEU. The atomic compositions of CsA-LEU-0.8HPMC-AB were much closer to those of CsA-LEU, indicating that the outer shell of the hollow particles was mainly composed of both CsA and LEU, while the inner layer was mainly composed of HPMC. To verify this deduction, the atomic compositions of CsA-0.8HPMC-AB without LEU were tested, which were much closer to those of CsA, indicating that the surface of CsA-0.8HPMC-AB was mainly covered by CsA, while the interior was mainly HPMC. Based on this, we could deduce that CsA-LEU-0.8HPMC-AB was the LHP composed of both CsA and LEU on the surface and HPMC on the inner layer.

### 3.6. Powder X-ray Diffraction

The PXRD results of the drug, excipients, and spray-dried powder formulations are shown in Figure 4. Raw CsA and HPMC had no representative crystal peaks but two broad peaks, indicating that raw CsA and HPMC had an amorphous structure [46,47]. The LEU had characteristic peaks with high intensities at 6.1°, 12.1°, 24.4°, 30.6°, and 36.9°, indicating that LEU had a crystalline structure [48,49]. The characteristic peaks of LEU could be observed on the spectra of CsA-LEU and CsA-LEU-0.8HPMC-AB with a much lower intensity, indicating that CsA-LEU and CsA-LEU-0.8HPMC-AB exhibited a low-crystalline structure after spray drying. Powders with an amorphous or low-crystalline structure are beneficial for poorly water-soluble drugs, as they could improve the dissolution rate and bioavailability [50]. 

### 3.7. Thermogravimetric Analysis

As shown in Figure 5A, the moisture content and thermal stability of the drug, excipients, and spray-dried powder formulations were analyzed by the TGA technique. All the supplied samples showed only a very slight weight loss (<4%) below 120 °C indicating the presence of a very small amount of moisture. The initial decomposition of CsA was from 244 °C to 334 °C with a weight loss of ~10%, followed by a sharp decomposition from 334 °C to 400 °C with a weight loss of ~86.5% due to the complex thermal degradation process. LEU showed a weight loss at 175 °C and 100% weight loss by 290 °C. The initial decomposition of HPMC was from 30 °C to 100 °C with a weight loss of ~3% due to water evaporation, and the second decomposition was from 240 °C to 325 °C with a weight loss of ~20%, followed by a significant weight loss up to 380 °C with ~13% residual weight. The initial decomposition of CsA-LEU was from 160 °C to 244 °C with a weight loss of ~17% due to the decomposition of LEU, which was consistent with the theoretical content of LEU with a value of 16.7% in the formulation, and the second decomposition was from 244 °C to 600 °C with a weight loss of ~83% due to the decomposition of CsA. CsA-LEU-0.8HPMC-AB exhibited similar TGA curves compared to CsA-LEU. The initial decomposition of CsA-LEU-0.8HPMC-AB was from 160 °C to 225 °C with a weight loss of ~13% due to the decomposition of LEU, which was proved by the theoretical content of LEU with a value of 12.5% in the formulation, followed by a weight loss of ~85.5% up to 600 °C due to the decomposition of CsA and HPMC. In addition, CsA-LEU-0.8HPMC-AB with a low moisture content (~0.25%) was beneficial for pulmonary delivery. The powders with amorphous or low-crystalline structures were metastable and tended to crystallize again at higher moisture content, and such changes might have a significant impact on the particle shape, size, dissolution, and aerosol performance of the DPIs [51,52]. Furthermore, the weight loss positions of the supplied samples were matched with their respective DSC curves.

### 3.8. Differential Scanning Calorimetry 

As shown in Figure 5B, the melting of the drug, excipients, and spray-dried powder formulations was analyzed by the DSC technique. Raw CsA exhibited an endothermic peak at 132.6 °C related to the glass transition temperature [53], followed by a broad endothermic peak at 260.2 °C for melting of partial crystalline form and decomposition at higher temperatures. LEU exhibited a sharp endothermic peak at 307.5 °C related to the melting point, reflecting its highly crystalline structure [54]. HPMC exhibited a broad endothermic peak at 82.1 °C related to the water loss event, with no melting peak when heating up to 320 °C, reflecting its amorphous nature [55]. CsA-LEU, composed of LEU and CsA, exhibited a broad endothermic peak from LEU, and the endothermic peaks of CsA were not observed. In addition, compared to LEU, the intensity of the endothermic peak of CsA-LEU decreased, and the peak shifted from 307.5 °C to 259.2 °C, indicating the crystalline structure was weakening. The DSC result of CsA-LEU-0.8HPMC-AB was similar to that of CsA-LEU, and the difference was that the intensity of the endothermic peak was much weaker and the endothermic peak shifted to a much lower temperature, indicating HPMC further loosened the crystal structure. In addition, as the concentration of HPMC increased in the formulation, the temperature of the endothermic peak showed a decreasing trend (Appendix A). The DSC results of the supplied samples were consistent with the PXRD results.

### 3.9. In Vitro Aerosolization Performance

The in vitro aerodynamic performances of the formulations were tested by NGI, and the deposition results and aerodynamic properties are shown in Figure 6 and Table 4, respectively. The deposition amounts of the formulations in the induction port and pre-separator were all less than 10%, indicating the particles were easily aerosolized and passed through the mouth and trachea, and the addition of AB in the formulation could further improve the aerodynamic performances of powders compared to formulations containing the same concentration of CsA, LEU, and HPMC, as evidenced by the deposition of more particles into the MOC (filter). The EF values of the formulations were all >84%, indicating good particle flowability and low adhesion to the capsule and inhaler. The FPF values of CsA-SD and CsA-LEU were 66.32 ± 0.96% and 75.44 ± 0.62%, respectively, indicating that LEU improved the in vitro particle deposition, which was attributed to LEU decreasing the cohesive and adhesive forces between particles to promote good aerosolization of the particles [56]. The high FPF values of the typical LHPs were in the range of 52% to 73%, which was related to the lower tap density [57]. In addition, for the typical LHPs, AB significantly improved the FPF value by 4–10% compared to formulations containing the same concentration of CsA, LEU, and HPMC, which was related to AB reducing the density of the formulation. The MMAD values, an important parameter attributed to the aerodynamic properties of respirable particles, were calculated to be in the range of 1.1 μm to 2.9 μm for the formulations. For the typical LHPs with or without AB, the MMAD values showed an increasing trend with the increase in HPMC concentration, and it could be calculated using Equation (1) to obtain the corresponding results. The geometric diameter of the LHPs was similar, while the tap density significantly increased with the increase in HPMC concentration; thus, the MMAD value also showed a corresponding increasing trend. In addition, the typical LHPs with AB showed a lower MMAD value compared to the typical LHPs without AB that contained the same concentration of CsA, LEU, and HPMC, related to the lower density. The LHPs CsA-LEU-0.8HPMC-AB had a minimum MMAD value of 1.15 ± 0.10 μm, indicating that CsA-LEU-0.8HPMC-AB were extra-fine particles (aerodynamic diameter less than 2 μm) and could be effectively deposited in the deeper lungs [58,59]. Based on this, CsA-LEU-0.8HPMC-AB was used for subsequent research.

### 3.10. Solubility Test

As shown in Figure 7, the solubility of CsA-LEU-0.8HPMC-AB in pure water was 36.71 μg/mL and was about 5.5-fold higher than that of raw CsA, indicating that CsA-LEU-0.8HPMC-AB significantly improved the solubility of CsA and predicted better dissolution properties. One of the most important reasons was that HPMC increased the solubility of the drug, which was attributed to intermolecular interactions between the drug and HPMC through hydrogen bonding. Using HPMC to increase the solubility of poorly water-soluble drugs has been reported in many publications [60,61,62], and the concentration of HPMC was a key factor affecting the solubility of the drug [63]. 

### 3.11. In Vitro Dissolution

The in vitro dissolution results of raw CsA and the CsA-LEU-0.8HPMC-AB formulation are shown in Figure 8. The dissolution results confirmed our predictions. Due to the poor water solubility of CsA, the release rate of raw CsA was very slow, and the release amount did not surpass 24% within 24 h. However, in the first 5 min, 49.66% of the CsA was released from CsA-LEU-0.8HPMC-AB, and all of the drug was released within 1 h. The improved dissolution performance might be attributed to the presence of HPMC in the particles. Similarly, HPMC was systematically evaluated to improve the dissolution rate of itraconazole succinic acid in the literature [63]. On the other hand, the rapid dissolution performance could also be related to the small particle size and large specific surface. In addition, the dissolved drug showed no significant recrystallization within 24 h, which might be explained by the following mechanisms. First, the dissolved HPMC could be adsorbed on the surface of the drug crystal through hydrophobicity and hydrogen bonding, reducing drug diffusion to the crystal surface [64]. Second, HPMC increased the solubility of the drug to nominally reduce the degree of supersaturation and inhibit drug crystallization [65]. Third, HPMC increased the viscosity of the dissolution medium, reducing diffusion and inhibiting crystallization [66]. Finally, dipalmitoylphosphatidylcholine, as a surfactant in the dissolution medium, inhibited aggregation, and LEU in the formulation reduced cohesion between the particles, which was also an important reason for improving the dissolution properties. The enhanced dissolution of the LHPs might be beneficial for improving their bioavailability in vivo.

### 3.12. In Vitro Cytotoxicity Assay

To investigate the biocompatibility and cytotoxicity of the LHPs, the cellular viability of CsA-LEU-0.8HPMC-AB was measured on A549 cells using an MTS assay. As shown in Figure 9, in the range of 1 to 100 μg/mL (equivalent to CsA concentration), the cell viability values were all greater than 90% within 24 h, and cell growth was not significantly inhibited. The results showed that CsA-LEU-0.8HPMC-AB as LHPs had favorable cytocompatibility and could safely deliver CsA via pulmonary inhalation administration.

### 3.13. In Vivo Pharmacokinetic Studies

CsA is one of the most important immunosuppressants in the field of organ transplantation; it is highly dependent on bile absorption and has a significant first-pass effect. Due to the poor solubility of CsA, its bioavailability after oral administration is not high. Excessive systemic exposure to CsA may cause serious side effects, especially nephrotoxicity and hepatotoxicity. The lungs have the advantages of a large surface area, good permeability, and low enzyme activity, making them particularly suitable for the delivery of therapeutic peptides and proteins [67]. Based on this, pharmacokinetic studies on oral administration of Neoral^®^ and inhalation administration of CsA-LEU-0.8HPMC-AB in rats were carried out to validate the advantage of the LHPs in this study, and the results are shown in Figure 10 and Table 5. The *C*_max_ and AUC_0–∞_ of CsA-LEU-0.8HPMC-AB were 109.7 ng/mL and 1662.0 ng·h/mL and increased by about 2-fold and 2.8-fold in contrast to Neoral^®^, respectively, with a significant difference (*p* < 0.05). The *T*_max_ and *T*_1/2_ of CsA-LEU-0.8HPMC-AB were 2.2 h and 7.6 h, respectively, with no significant difference in contrast to Neoral^®^ (*p* > 0.05). The superior *C*_max_ and AUC_0–∞_ values indicated that inhalation administration of the LHPs could significantly improve the in vivo bioavailability of CsA compared with oral administration of Neoral^®^, which was due to the extra-fine particles, enhanced solubility, and enhanced dissolution of the LHPs. The LHPs, with the advantage of avoiding alveolar macrophage clearance due to their large geometric size, could effectively deliver CsA to the systemic circulation by inhalation administration to achieve therapeutic drug concentrations with lower systemic exposure than oral administration and could be a viable delivery option for CsA.

The therapeutic range of CsA is narrow in clinical therapy, and the concentration of CsA in plasma needs to be controlled within 50 to 300 ng/mL [68]. Therefore, further systematic evaluation is needed for the dosage and frequency of inhalation administration of LHPs in clinical therapy. On the other hand, LEU and HPMC are commonly used excipients in the preparation of DPIs [69], but they have not yet entered the list of acceptable excipients for pulmonary delivery. LHPs show no significant cytotoxicity in vitro, but the inhalation toxicity of chronic use is still uncertain. Consequently, the safety and tolerability of LHPs for pulmonary delivery are required to be systematically evaluated before clinical use.

## 4. Conclusions

In this study, typical LHPs were successfully prepared for the pulmonary delivery of CsA. The typical LHPs were spherical particles composed of both CsA and LEU on the surface and HPMC on the inner layer, with a geometric diameter (D_90_) of 10–13 μm, a tap density of 42.03–111.58 mg/cm^3^, and a Carr’s index of 27–32. As a typical LHP, CsA-LEU-0.8HPMC-AB showed excellent in vitro aerodynamic performance, with a minimum MMAD value of 1.15 μm, an EF value of 86.79%, and an FPF value of 68.04%, indicating that CsA-LEU-0.8HPMC-AB could be effectively deposited in the deeper lungs. The solubility of CsA-LEU-0.8HPMC-AB in pure water was about 5.5-fold higher than that of raw CsA, and the dissolution of CsA-LEU-0.8HPMC-AB suggested that all of the drug was released within 1 h. In addition, in vivo pharmacokinetics results suggested that inhalation administration of CsA-LEU-0.8HPMC-AB could significantly improve the in vivo bioavailability of CsA compared with oral administration of Neoral^®^. In brief, these findings indicated that the LHPs could effectively deliver CsA to the systemic circulation by inhalation administration with lower systemic exposure than oral administration and could be a viable delivery option for CsA.

## Figures and Tables

**Figure 1 pharmaceutics-15-02204-f001:**
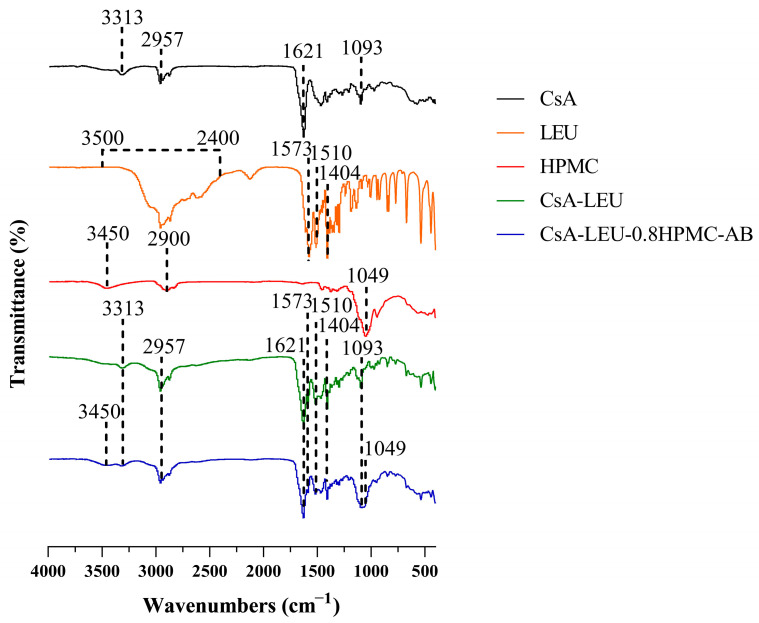
FTIR spectra of raw CsA, LEU, HPMC, and CsA-LEU and CsA-LEU-0.8HPMC-AB formulations.

**Figure 2 pharmaceutics-15-02204-f002:**
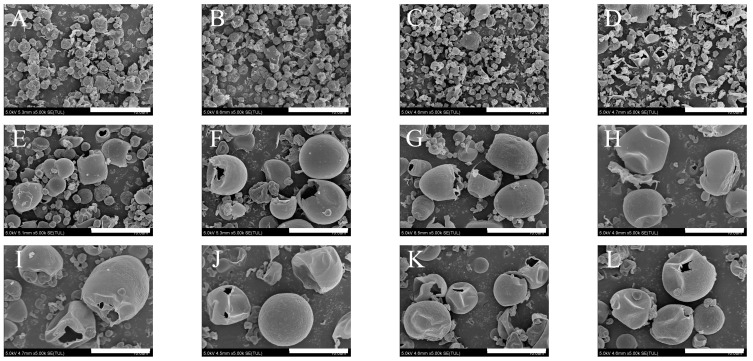
SEM images of the formulations: (**A**) CsA-SD; (**B**) CsA-LEU; (**C**) CsA-LEU-0.4HPMC; (**D**) CsA-LEU-0.4HPMC-AB; (**E**) CsA-LEU-0.8HPMC; (**F**) CsA-LEU-0.8HPMC-AB; (**G**) CsA-LEU-1.2HPMC; (**H**) CsA-LEU-1.2HPMC-AB; (**I**) CsA-LEU-1.6HPMC; (**J**) CsA-LEU-1.6HPMC-AB; (**K**) CsA-LEU-2.0HPMC; (**L**) CsA-LEU-2.0HPMC-AB. Scale bar: 10.0 μm.

**Figure 3 pharmaceutics-15-02204-f003:**
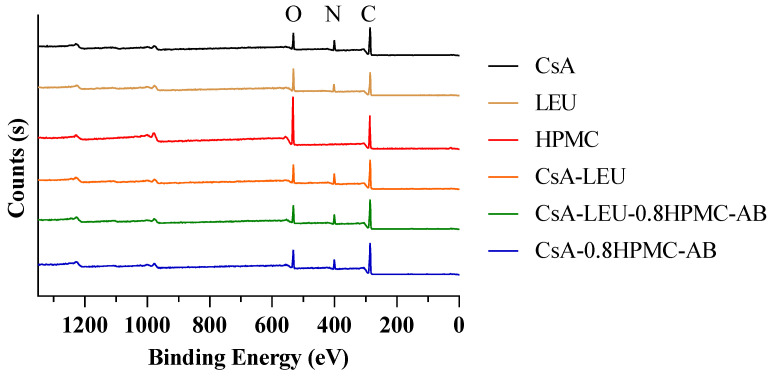
XPS spectra of CsA, LEU, HPMC, and CsA-LEU, CsA-LEU-0.8HPMC-AB, and CsA-0.8HPMC-AB. CsA-0.8HPMC-AB was the formulation with 2 mg/mL CsA, 0.8 mg/mL HPMC, and 2 mg/mL AB in the spray drying process.

**Figure 4 pharmaceutics-15-02204-f004:**
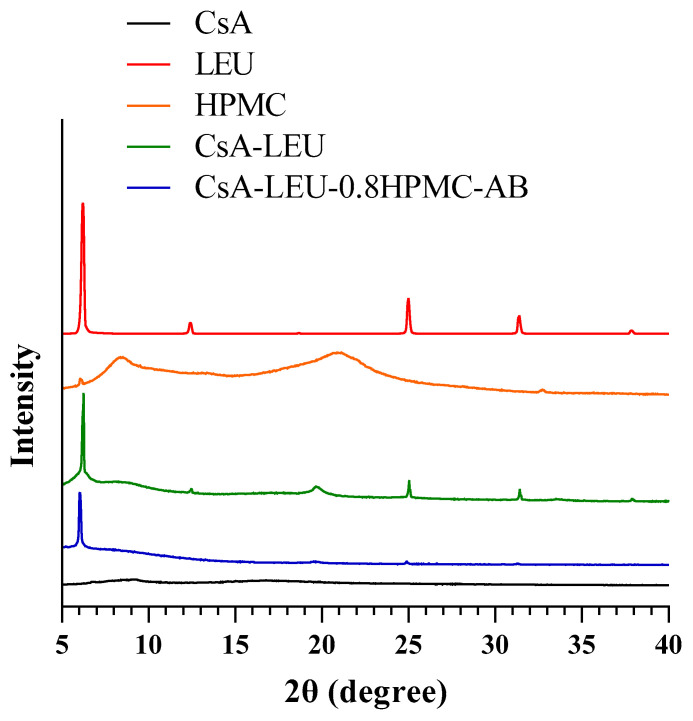
PXRD results of CsA, LEU, HPMC, and CsA-LEU and CsA-LEU-0.8HPMC-AB formulations.

**Figure 5 pharmaceutics-15-02204-f005:**
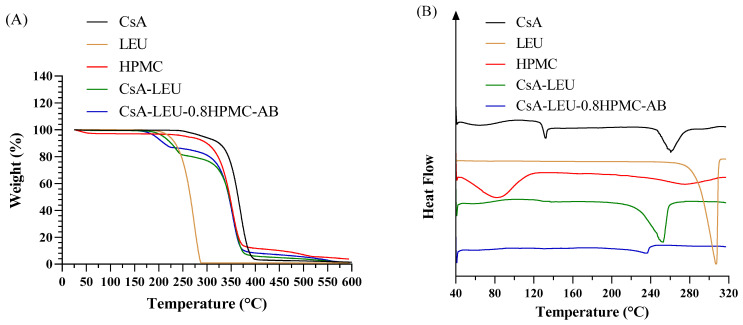
(**A**) TGA and (**B**) DSC curves of CsA, LEU, HPMC, and CsA-LEU and CsA-LEU-0.8HPMC-AB formulations.

**Figure 6 pharmaceutics-15-02204-f006:**
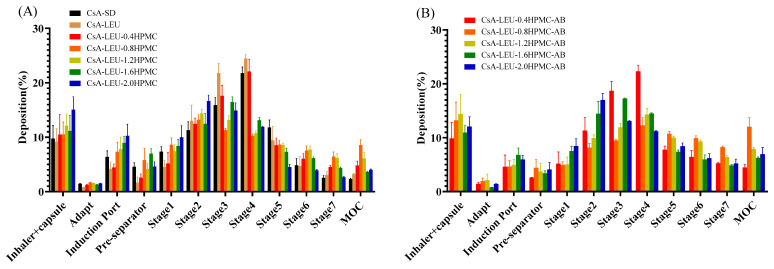
In vitro aerodynamic deposition of formulations at a flow rate of 90 L/min by NGI: (**A**) formulations without AB; (**B**) formulations with AB. Data are expressed as the mean ± SD (*n* = 3).

**Figure 7 pharmaceutics-15-02204-f007:**
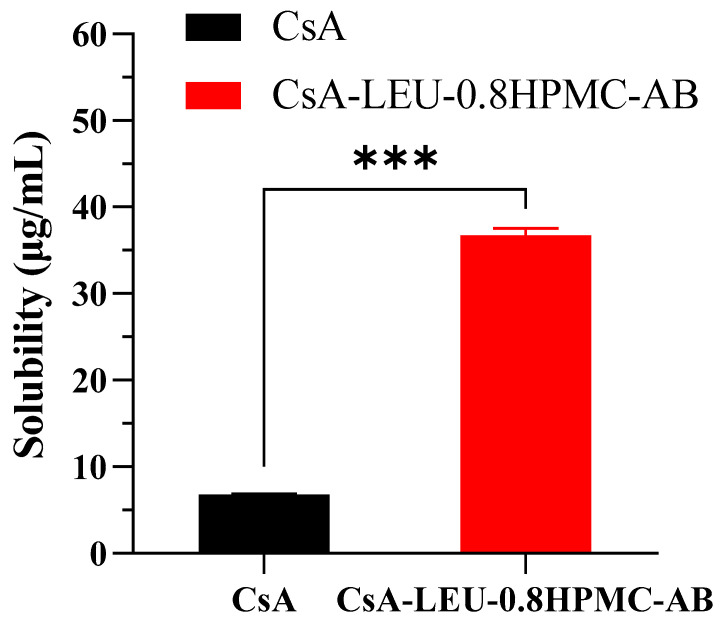
Solubility results of raw CsA and the formulation CsA-LEU-0.8HPMC-AB. Data are expressed as the mean ± SD (*n* = 3). Level of significance: *** *p* < 0.001.

**Figure 8 pharmaceutics-15-02204-f008:**
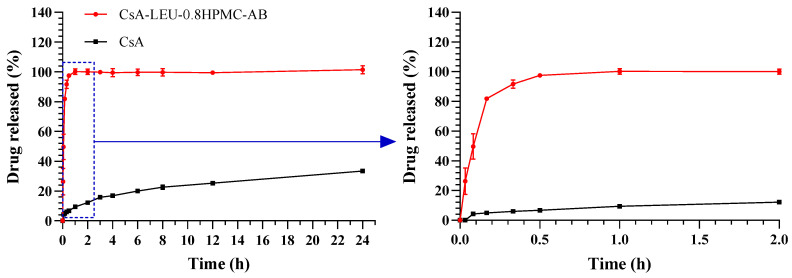
In vitro dissolution results of raw CsA and the CsA-LEU-0.8HPMC-AB formulation. Data are expressed as the mean ± SD (*n* = 3).

**Figure 9 pharmaceutics-15-02204-f009:**
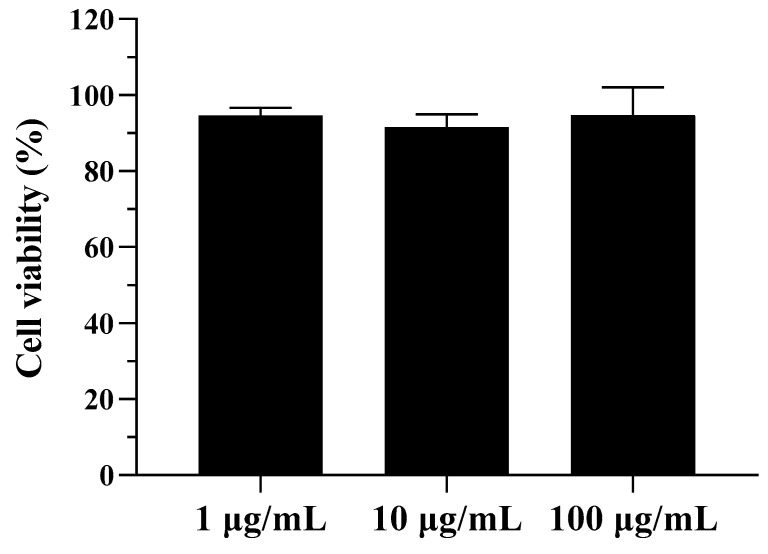
Cell viability of CsA-LEU-0.8HPMC-AB in the A549 cell line. After a 24 h incubation period, an MTS assay was performed to check the cell replication. Data are expressed as the mean ± SD (*n* = 3).

**Figure 10 pharmaceutics-15-02204-f010:**
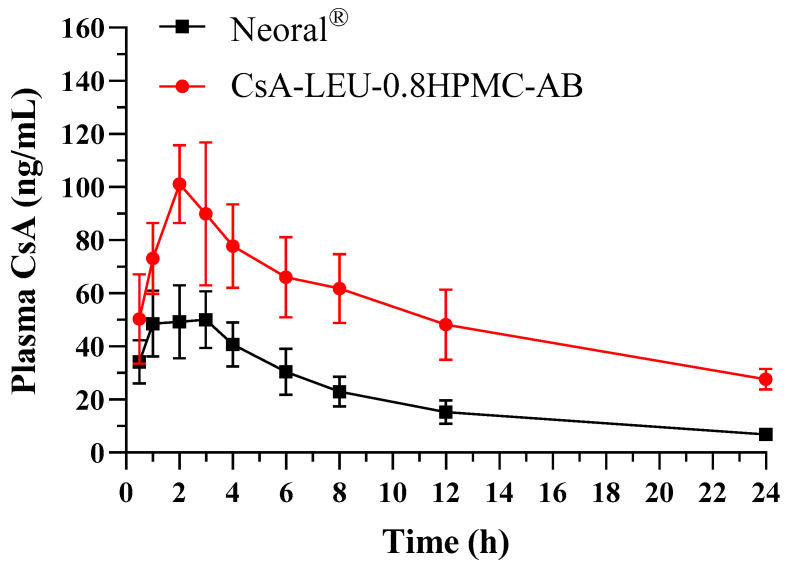
Concentration–time curve of CsA in rats after oral administration of Neoral^®^ (0.4 mg CsA/kg) and inhalation administration of CsA-LEU-0.8HPMC-AB (0.4 mg CsA/kg). Data are expressed as the mean ± SD (*n* = 6).

**Table 1 pharmaceutics-15-02204-t001:** Composition of the solutions for spray drying.

Formulation	CsA (mg/mL)	LEU (mg/mL)	HPMC (mg/mL)	AB (mg/mL)
CsA-SD	2	0	0	0
CsA-LEU	2	0.4	0	0
CsA-LEU-0.4HPMC	2	0.4	0.4	0
CsA-LEU-0.4HPMC-AB	2	0.4	0.4	2
CsA-LEU-0.8HPMC	2	0.4	0.8	0
CsA-LEU-0.8HPMC-AB	2	0.4	0.8	2
CsA-LEU-1.2HPMC	2	0.4	1.2	0
CsA-LEU-1.2HPMC-AB	2	0.4	1.2	2
CsA-LEU-1.6HPMC	2	0.4	1.6	0
CsA-LEU-1.6HPMC-AB	2	0.4	1.6	2
CsA-LEU-2.0HPMC	2	0.4	2.0	0
CsA-LEU-2.0HPMC-AB	2	0.4	2.0	2

**Table 2 pharmaceutics-15-02204-t002:** Content, size distribution (D_90_), density properties (bulk and tap densities), and flowability parameters (Carr’s index) of the formulations. Data are expressed as the mean ± SD (*n* = 3).

Formulation	Theoretical Content (%)	Content (%)	D_90_ (µm)	Bulk Density (mg/cm^3^)	Tap Density (mg/cm^3^)	Carr’s Index (%)
CsA-SD	-	-	4.51 ± 0.03	115.13 ± 2.00	210.67 ± 3.21	45.33 ± 1.44
CsA-LEU	83.33	83.86 ± 0.13	4.88 ± 0.03	124.53 ± 3.31	259.47 ± 0.87	52.00 ± 1.44
CsA-LEU-0.4HPMC	71.43	71.74 ± 0.18	8.37 ± 0.13	95.09 ± 2.74	136.25 ± 1.81	30.22 ± 1.10
CsA-LEU-0.4HPMC-AB	71.43	71.84 ± 0.15	8.61 ± 0.08	68.95 ± 2.52	95.56 ± 2.44	27.87 ± 0.82
CsA-LEU-0.8HPMC	62.50	62.40 ± 0.29	10.97 ± 0.05	45.53 ± 1.65	63.69 ± 1.40	28.53 ± 1.05
CsA-LEU-0.8HPMC-AB	62.50	61.96 ± 0.17	11.57 ± 0.05	30.40 ± 2.16	42.03 ± 2.14	27.73 ± 1.68
CsA-LEU-1.2HPMC	55.56	55.05 ± 0.22	10.87 ± 0.05	49.85 ± 1.69	73.30 ± 1.87	32.00 ± 0.65
CsA-LEU-1.2HPMC-AB	55.56	55.45 ± 0.27	11.90 ± 0.08	42.04 ± 1.49	58.38 ± 1.56	28.00 ± 0.65
CsA-LEU-1.6HPMC	50.00	50.21 ± 0.25	10.87 ± 0.12	59.92 ± 2.17	86.27 ± 3.23	30.53 ± 0.75
CsA-LEU-1.6HPMC-AB	50.00	49.99 ± 0.18	11.40 ± 0.14	45.67 ± 1.13	65.48 ± 1.46	30.27 ± 0.19
CsA-LEU-2.0HPMC	45.45	45.23 ± 0.10	11.03 ± 0.12	77.80 ± 1.01	111.58 ± 2.02	30.27 ± 0.38
CsA-LEU-2.0HPMC-AB	45.45	45.17 ± 0.16	12.43 ± 0.05	48.68 ± 2.09	68.87 ± 2.49	29.33 ± 0.50

**Table 3 pharmaceutics-15-02204-t003:** Percentage atomic compositions of the formulations determined from XPS analysis (Sto: stoichiometric).

Formulation	C	N	O
CsA (Sto)	72.94	12.94	14.12
LEU (Sto)	66.67	11.11	22.22
HPMC (Sto)	60.00	0	40.00
CsA (XPS)	75.40	11.96	12.64
LEU (XPS)	68.64	11.16	20.20
HPMC (XPS)	66.39	0	33.61
CsA-LEU (XPS)	73.12	11.41	15.47
CsA-LEU-0.8HPMC-AB (XPS)	72.54	12.10	15.36
CsA-0.8HPMC-AB (XPS)	75.01	11.64	13.36

**Table 4 pharmaceutics-15-02204-t004:** In vitro aerodynamic properties of formulations: emitted fraction (EF), fine particle fraction (FPF), and mass median aerodynamic diameter (MMAD). Data are expressed as the mean ± SD (*n* = 3).

Formulation	EF (%)	FPF < 5 μm (%)	MMAD (μm)
CsA-SD	90.24 ± 1.97	66.32 ± 0.96	2.09 ± 0.10
CsA-LEU	90.81 ± 1.94	75.44 ± 0.62	2.18 ± 0.12
CsA-LEU-0.4HPMC	89.51 ± 3.04	71.86 ± 2.40	2.04 ± 0.05
CsA-LEU-0.8HPMC	89.48 ± 1.87	61.11 ± 2.74	1.88 ± 0.21
CsA-LEU-1.2HPMC	87.89 ± 1.76	61.76 ± 1.76	2.08 ± 0.08
CsA-LEU-1.6HPMC	88.83 ± 2.28	58.69 ± 0.22	2.31 ± 0.07
CsA-LEU-2.0HPMC	84.94 ± 1.94	52.24 ± 2.10	2.87 ± 0.05
CsA-LEU-0.4HPMC-AB	90.11 ± 2.42	72.53 ± 1.08	2.04 ± 0.08
CsA-LEU-0.8HPMC-AB	86.79 ± 2.76	68.04 ± 0.93	1.15 ± 0.10
CsA-LEU-1.2HPMC-AB	85.61 ± 2.98	66.31 ± 1.20	1.56 ± 0.02
CsA-LEU-1.6HPMC-AB	89.03 ± 1.02	65.37 ± 0.61	2.39 ± 0.02
CsA-LEU-2.0HPMC-AB	87.92 ± 1.47	61.87 ± 1.24	2.32 ± 0.09

**Table 5 pharmaceutics-15-02204-t005:** Pharmacokinetic parameters of CsA formulations. *C*_max_: maximum concentration; *T*_max_: time to maximum concentration; AUC_0–∞_: area under the curve of blood concentration vs. time from 0 h to ∞ h. Data are expressed as the mean ± SD (*n* = 6).

Formulation	*C*_max_ (ng/mL)	*T*_max_ (h)	*T*_1/2_ (h)	AUC_0–∞_ (ng·h/mL)
Neoral^®^	54.4 ± 10.1	2.2 ± 0.8	7.1 ± 2.5	585.2 ± 98.0
CsA-LEU-0.8HPMC-AB	109.7 ± 18.9 ***	2.2 ± 0.4	7.6 ± 2.7	1662.0 ± 382.1 ***

*** significant difference with *p* < 0.001 compared with Neoral^®^.

## Data Availability

The data that support the findings of this study are available from the corresponding author upon reasonable request.

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
