# Peer review of "Development of Large Hollow Particles for Pulmonary Delivery of Cyclosporine A"

_pharmaceutics, 2023, doi:10.3390/pharmaceutics15092204_

Round 1

Reviewer 1 Report

The current research article is well designed, and results are well documented. The prepared LHPs of CsA-LEU-0.8HPMC-AB are significantly bioavailable compared to oral formulation of Neoral. The authors have done all the physicochemical characterizations which is strengthening its point. However, there are some doubts that must be cleared up and a few modifications must be made before accepting the manuscript.

Some results described in result sections are just the figure explanations. Authors haven’t done the critical analysis of results. It will be nice if they discuss the results with proper citations. E.g. in section Results 3.6 please mention what is the significance of “low crystalline” structure of LHP in nasal delivery or formulation.

Why did the authors not compare the nasal pharmacokinetics of CsA-LEU-0.8HPMC-AB with any marketed nasal formulation of CsA?

In Figure 1. FTIR, please annotate the important peaks (along with functional groups) which are shifted, in such way figure itself will become explanatory.  

Minor editing of English language required

Reviewer 2 Report

Major comments:

1. What was the primary rationale to select CsA concentrations considering a the main aim of the study? Did authors carry out a pilot study assessing range of different concentrations of experimental nanocompounds/particles to validate the null hypothesis? 

2. Why authors decided to use A549 cell line for cytotoxicity assay, not a more relevant, standardized cell line for such research related to immunosuppressive drugs?  

3. No XPS spectra presented, only reference to supplementary figure S2, not included in manuscript

4. All SEM microphotographs have the same scale bar 10um, despite obvious, different magnifications in 1st row of sub-figures vs 2nd/3rd rows. Explanation needed.  

5. Thermogravimetric analysis does not seem an essential part of methods applied as it has not delivered any added value overall.  The authors used a wide range of various methods/analyses without providing a detailed scientific justification.  

6. Constraints associated with XPS superficial chemical characterisation need to be described, as XPS analysis may not clearly reflect a complex ultrastructure of investigated nanocompounds. 

7. Broader discussion describing inhaled CsA-based nanoparticles pharmacokinetics is recommended, considering hypothetical clinical settings. 

5. Missing declaration of compliance with ethical recommendations addressing animal studies. 

Minor revisions required to comply with plain scientific English standards.
